# Plasticity in Gene Expression Patterns and *CYPSF* Gene Possibly Involved in the Etofenprox-Resistant Population of White-Backed Planthopper, *Sogatella furcifera*

**DOI:** 10.3390/ijms252413605

**Published:** 2024-12-19

**Authors:** Murtaza Khan, Changhee Han, Minyoung Choi, Hoki Hong, Nakjung Choi, Juil Kim

**Affiliations:** 1Agriculture and Life Sciences Research Institute, Kangwon National University, Chuncheon 24341, Republic of Korea; murtazakhan@kangwon.ac.kr; 2Interdisciplinary Graduate Program in Smart Agriculture, Kangwon National University, Chuncheon 24341, Republic of Korea; 201711031@kangwon.ac.kr; 3Department of Plant Medicine, Division of Bio-Resource Sciences, College of Agriculture and Life Science, Kangwon National University, Chuncheon 24341, Republic of Korea; alsdudcom@kangwon.ac.kr (M.C.); knuduck@kangwon.ac.kr (H.H.); 4Crop Foundation Research Division, National Institute of Crop Science, Rural Development Administration, Wanju 55365, Republic of Korea; njchoi@korea.kr

**Keywords:** white-backed planthopper, insecticide resistance, RNA-seq analysis, mutation, metabolic resistance

## Abstract

The white-backed planthopper (WBPH) poses a significant threat to rice crops globally. A bioassay was conducted on three WBPH populations collected from Korean rice fields to assess the effectiveness of five insecticides, including etofenprox and fenobucarb. The results showed a mortality rate of over 97% at the recommended concentration for carbamate and organophosphate insecticides. However, etofenprox exhibited a mortality rate of less than 40% in all tested populations with the Jindo population showing the highest resistance. No mutations were identified in the voltage-sensitive sodium channel, the target site of etofenprox, suggesting an alternative resistance mechanism. To explore this, RNA-seq analysis was performed on the Jindo population to identify genes potentially associated with etofenprox resistance. Gene expression was assessed after treatment with two sublethal doses of etofenprox using the Jindo population. The analysis revealed that the *CYPSF* gene, part of the CYP6 family, was consistently overexpressed in both treated and untreated samples. This observation aligns with the bioassay results, where mortality increased significantly after treatment with the cytochrome P450 inhibitor PBO, indicating that CYPSF may play a key role in etofenprox resistance. Additionally, distinct gene expression patterns at different etofenprox concentrations suggest that metabolic resistance mechanisms may be involved.

## 1. Introduction

Rice is the primary food source for over three billion people globally, making it vital for food security in many countries, including Korea [1,2]. However, various biological and environmental factors frequently affect rice production; one major factor is insects, which can result in significant yield losses and a drop in grain quality [3,4]. The white-backed planthopper (WBPH), *Sogatella furcifera* (Horváth) (Hemiptera: Delphacidae), is a sucking pest that causes significant harm to rice crops and is one of the most devastating pests in areas where rice is grown [5].

WBPH migrates from the subtropical to temperate regions, arriving in Korea from neighboring countries almost yearly [6]. Because of its severe damaging effects, WBPH has been included in Korea’s National Species List [7]. This pest damages rice plants by feeding on them, lowering the weight and percentage of kernel filling, causing hopper burn, and spreading viruses like the southern rice black-streaked dwarf virus (SRBSDV), which can result in substantial yield losses [8,9,10].

Chemical control is still the most widely used and successful strategy for controlling WBPH and stopping the spread of SRBSDV transmission in paddy fields [11]. Several insecticides, such as organophosphates, carbamates, pyrethroids, insect growth regulators, and neonicotinoids, have been used to alleviate the effects of WBPH [12]. However, the excessive and improper use of these chemicals has led to significant resistance in WBPH and other insect pests, posing a major challenge for pest management [12,13].

Insects develop insecticide resistance through several mechanisms, including mutations at the target site and activating detoxification enzymes [4,14]. The primary mechanism underlying the development of insecticide resistance is metabolic resistance, which is achieved by increasing the activities of detoxification enzymes such as cytochrome P450 monooxygenase (CYP), esterase (EST), and glutathione S-transferase (GST) [15]. Additionally, other detoxifying enzymes, such as ATP-binding cassette (ABC) transporters, UDP-glycosyltransferase (UGT), and cuticular proteins (CP), also play significant roles in enhancing insecticide resistance [16,17,18]. Studies have shown that WBPH exhibits an increased expression of *P450*, *carboxylesterase precursor* (*EST-1*), and *GST* in response to various insecticides, such as neonicotinoids, insect growth regulators, and organophosphates, contributing to its resistance [19,20,21].

Furthermore, WBPH has been found to have high resistance to buprofezin, chlorpyrifos, and pymetrozine, and low to moderate resistance to nitenpyram, clothianidin, thiamethoxam, imidacloprid, dinotefuran, isoprocarb, and etofenprox [5]. The authors also noted that if used in low concentrations, pymetrozine, isoprocarb, and etofenprox were at risk of failure. The upregulation of *CYP* genes has been closely associated with this resistance, as seen with the increased activity of CYP enzymes and the expression of specific *CYP* genes in response to insecticide exposure [22,23].

Pyrethroids are synthetic insecticides commonly used to control various rice pests. They target the voltage-gated sodium channels in insect nerve cells, disrupting normal nerve function and leading to insect death [24,25]. However, insects have developed resistance to pyrethroids through knockdown resistance, which involves mutations in sodium channels and enhanced metabolic detoxification, mainly through increased activity of CYP [26,27].

In the current study, we investigated the resistance levels of WBPH to five different insecticides, including etofenprox (3A), fenobucarb (1A), chlorpyrifos-methyl (1B), phenthoate (1B), and acephate (1B) across three regions of Korea (Sinan, Hadong, and Jindo) in 2019. Etofenprox, in particular, is widely used for planthopper management due to its broad-spectrum activity and relatively low environmental impact. However, resistance development threatens its efficacy, posing challenges to sustainable pest control. To address this concern, we further investigated the resistance of the WBPH population from Jindo to various concentrations (0 ppm, 25 ppm, and 100 ppm) to assess its sublethal effects and further explored potential resistance mechanisms. These include target site mutations and metabolic resistance mechanisms with a particular focus on detoxification-related genes.

Additionally, this study examines the role of gene expression plasticity in contributing to metabolic resistance. Here, we define plasticity as the ability of WBPH populations to exhibit dynamic gene expression changes in response to insecticide exposure, potentially enabling detoxification and resistance without permanent genetic alterations. This approach emphasizes functional gene expression changes rather than the evolutionary concept of plasticity as a heritable trait [28], providing insights into the adaptive responses of WBPH to sublethal doses of etofenprox.

## 2. Results

### 2.1. Bioassay Results

Rice seedling dip method-based bioassays were conducted to evaluate the resistance levels of WBPH populations collected from three locations in Korea (Sinan, Hadong, and Jindo) to five different insecticides. According to our research, exposure to fenobucarb (1A: carbamate), chlorpyrifos-methyl (1B: organophosphate), phenthoate (1B: organophosphate), and acephate (1B: organophosphate) resulted in an over 97% mortality rate for all WBPH populations. These clarifications refer to the insecticide’s modes of action (MoA) as defined by the Insecticide Resistance Action Committee. Nonetheless, there were differences in the population’s reactions to etofenprox (3A). The WBPH population from Hadong exhibited a mortality rate of 42.4 ± 5.2%, the population from Sinan showed a mortality rate of 40.0 ± 10.0%, and the population from Jindo had a mortality rate of 27.3 ± 9.1% (Table 1). Although it is challenging to evaluate resistance rates due to the absence of WBPH-susceptible strains for etofenprox, the bioassay results clearly indicate that the control efficiency of etofenprox at the recommended concentration was low for field populations.

To confirm the resistance observed in the Jindo population, we determined the LC_50_ value for etofenprox in this population. The findings showed that etofenprox alone had an LC_50_ value of 241.7 ppm with a 95% confidence interval (C.I.) of 157.69 to 373.28 ppm. Etofenprox’s LC_50_ value notably dropped to 51.0 ppm with a 95% C.I. of 10.6 to 144.7 ppm when it was coupled with 0.5% piperonyl butoxide (PBO). Etofenprox alone had a slope of 1.15 and a chi-square value of 3.42, but the combined treatment had a slope of 0.58 and a chi-square value of 1.81. Furthermore, the susceptibility resistance (SR) was 4.7 (Table 2).

These findings imply that whereas WBPH populations demonstrate total mortality when exposed to fenobucarb, chlorpyrifos-methyl, phenthoate, and acephate, they are resistant to etofenprox, especially in the Jindo WBPH population. The reported resistance in the Jindo WBPH population is likely caused by metabolic resistance mechanisms, most likely involving detoxification enzymes, as seen by the considerable drop in LC_50_ observed when etofenprox is coupled with PBO.

### 2.2. Mutation Survey in the Voltage-Gated Sodium Channel Gene

To identify mutations in the voltage-gated sodium channel, which is well known as the primary mechanism of pyrethroid insecticides for various sucking pests, including brown planthopper (*Nilaparvata lugens*), the sequence of the full-length gene was analyzed using the Jindo population of WBPH. The 6422 bp sodium channel gene (PQ306230.1) encoding 2069 amino acids matched the reference genome (GCA_017141385.1). It was confirmed to consist of 37 exons (Figure 1A). Five major mutations (V253F, T267A, M918T, L1014F, and F1534C) reported in sucking pests, including *N. lugens* [29], were investigated in the Jindo population with the highest level of etofenprox resistance, but no mutations were found (Figure 1B). Since mutation confirmation was based on pooled gDNA from a population of 20 adults, not at the individual level, it was confirmed that there was no mutation within the population. The information about the used primers are listed in Appendix A.

### 2.3. Raw and Trimmed Data Statistics of RNA-Seq

To explore the molecular mechanisms underlying etofenprox resistance, RNA-seq was conducted on three distinct WBPH populations: JD_100ppm (treated with 100 ppm etofenprox), JD_25ppm (treated with 25 ppm etofenprox), and JD_NT (untreated control), each with three biological replicates. The quality of the raw sequencing data was assessed by determining total read bases, total reads, GC content (%), and Q30 (%) values across the nine samples. The total read bases ranged from 11.35 Gb to 14.60 Gb with total reads varying between 112.38 million and 144.53 million. GC content ranged from 41.88% to 44.12%, while Q30 values consistently exceeded 95%, indicating high-quality data across all samples (Appendix A). These metrics confirm the reliability of the raw sequencing data for further analysis.

After processing the raw data using the Trimmomatic tool to remove adapter sequences and low-quality bases, the total read bases per sample ranged from 11.14 Gb to 14.33 Gb, and total reads varied from 110.86 million to 142.49 million. The GC content remained stable, ranging from 41.90% to 44.12%. Q30 values showed a slight improvement, with values consistently reaching or exceeding 96%, demonstrating further enhancement of data quality after trimming (Appendix A). These high-quality trimmed data are well suited for reliable downstream genetic analyses.

### 2.4. De Novo Assembly of Unigene Sets

To accurately represent the gene expression profiles in the WBPH populations and ensure comprehensive transcriptome reconstruction, we performed de novo assembly following the preprocessing of RNA-seq data. This assembly aimed to minimize errors and provide a reliable dataset for gene expression analysis. The statistics for the initial assembled contig, longest contig, and unigene contig are detailed in Table 3, which highlights key metrics such as the number of genes, number of transcripts, GC content (%), N_50_ values, average contig length (bp), and total assembled bases.

For the initial assembled contig, the dataset contained 222,775 genes and 294,336 transcripts with a GC content of 39.82%. The N_50_ value, a metric indicating the quality of the assembly, was 1330 bp, and the average contig length was 745.45 bp. The total assembled bases amounted to 219,415,085 bp, indicating the overall size and completeness of the assembly.

In the longest contig assembly, the number of genes matched the initial assembly at 222,775, but the number of transcripts was lower with only 222,775 transcripts recorded. The GC content was slightly lower at 39.43%, and the N_50_ value decreased to 966 bp with an average contig length of 629.77 bp. The total assembled bases for this assembly were 140,297,723 bp, reflecting a smaller total dataset.

The unigene contig assembly, which aimed to consolidate unique gene sequences, included 194,501 genes and transcripts. The GC content was the lowest among the three assemblies, at 39.38%. However, the N_50_ value was relatively high, at 1080 bp, indicating improved assembly contiguity compared to the longest contig. The average contig length was 660.27 bp with a total assembled base count of 128,423,880 bp (Table 3).

These assembly results underscore the robustness of the transcriptome data, ensuring its suitability for further functional and gene expression analyses.

### 2.5. ORF Prediction

Out of the 194,501 unigenes compiled for the open reading frame (ORF) prediction analysis (Table 4), 27,079 unigenes (13.92%) were found to contain predicted ORFs. Among these, the majority (24,705 unigenes, or 91.23%) contained a single predicted ORF, while a smaller portion (2374 unigenes, or 8.77%) contained multiple predicted ORFs. These findings indicate that a substantial portion of the assembled unigens may code for proteins with most containing a single ORF.

This ORF prediction analysis highlights the coding potential and functional diversity within the assembled unigene dataset, emphasizing the predominance of unigenes with single ORFs.

### 2.6. DEG Analysis in WBPH Populations Exposed to Etofenprox

The RNA-seq data from all samples were analyzed using the merged reference unigene set to evaluate gene expression and identify differentially expressed genes (DEGs). A key aspect of this analysis involved determining the mapping efficiency of reads to the reference unigene set, which helps assess the quality and completeness of the unigene assembly. The mapping statistics for all samples are presented in Appendix A, showing the number of processed reads, mapped reads, and unmapped reads for each sample.

The results revealed a high overall mapping ratio across all samples with mapping efficiencies ranging from 77.52% to 80.75%. For instance, JD_100ppm-1 had 131,877,316 processed reads, of which 105,239,130 reads (79.8%) were mapped, leaving 26,638,186 reads (20.2%) unmapped. Similarly, JD_100ppm-2 and JD_100ppm-3 had mapping efficiencies of 78.6% and 80.75%, respectively, with similar proportions of unmapped reads.

The mapping efficiencies for the JD_25ppm and JD_NT samples ranged from 77.52% to 80.22%, indicating a consistently high alignment of the sequence reads to the unigene set. Despite this, a noticeable proportion of unmapped reads remained, with values between 19.25% and 22.48%, depending on the sample (Appendix A).

These mapping ratios suggest that while the unigene assembly captured the majority of the sequence reads, a subset of the reads remained unmapped, which was potentially due to sequence variants, novel genes, or incomplete assembly. Nonetheless, the high mapping efficiency supports the comprehensiveness of the unigene set, ensuring robust data for DEG analysis.

We investigated the sublethal effects of etofenprox on the Jindo WBPH population by conducting a heat map analysis, principal component analysis plot (PCA), and MA plot analysis after the bioassay analysis. We compared three groups: Jindo BPH susceptible which was not treated with etofenprox (JD_NT), Jindo WBPH treated with 25 ppm etofenprox (JD_25ppm), and Jindo WBPH treated with 100 ppm etofenprox (JD_100ppm). The objective was to assess the sublethal effects of etofenoprox at various concentrations (0 ppm, 25 ppm, and 100 ppm) on the Jindo WBPH population. The heat map and PCA analysis revealed that JD_25ppm displayed a distinct resistance profile compared with JD_NT and JD_100ppm (Figure 2A,B). Interestingly, JD_100ppm and JD_NT did not significantly differ in resistance, indicating that increased etofenprox dosage may not always result in increased resistance in the investigated scenarios (Figure 2).

We conducted an MA plot analysis to further corroborate these findings by visualizing the gene expression patterns between the different treatment groups. An MA plot (Mean-Average plot) is a graphical representation widely used in transcriptomics to display differential gene expression. It plots the log-fold change (M, representing the magnitude of expression differences) on the *Y*-axis against the average expression levels (A, representing the mean expression of each gene across conditions) on the *X*-axis. This visualization highlights upregulated and downregulated genes while accounting for their overall expression levels.

In the comparison between JD_25ppm and JD_NT, 1501 genes were upregulated while 1208 genes were downregulated, indicating significant transcriptional changes associated with moderate etofenprox resistance (Figure 2C1). In the comparison between JD_100ppm and JD_NT, 1188 genes were upregulated, and 1150 genes were down-regulated, suggesting a complex but somewhat less pronounced transcriptional response to the higher dose of etofenprox (Figure 2C2). Finally, when comparing JD_100ppm to JD_25ppm, we observed 1203 genes upregulated and 1424 genes downregulated, indicating significant differences in gene expression between these two resistance levels (Figure 2C3).

These results indicate that compared to higher exposure levels (100 ppm), etofenprox exposure (25 ppm) triggers a more prominent resistance mechanism. This could suggest that the detoxification or resistance pathways have reached a saturation threshold. The notable alterations in gene expression, particularly in JD_25ppm, emphasize the intricacy of resistance mechanisms and stress the significance of taking dosage and exposure length into account when evaluating pesticide resistance. These results advance our knowledge of the molecular responses of WBPH populations to varying etofenprox concentrations (0 ppm, 25 ppm, and 100 ppm), which will aid in creating more potent resistance management techniques.

### 2.7. Plasticity of Detoxification Enzyme Gene Expression

Phenotypic plasticity refers to the ability of a single genotype to produce different phenotypes in response to varying environmental conditions [28]. In this study, we examined the plasticity of detoxification enzyme gene expression to understand how sublethal doses of etofenprox influence transcriptional responses in WBPH populations collected from Jindo (JD), Korea. While we did not directly assess the genetic uniformity among treatment groups, all populations used in this study originated from a single field population and were reared under controlled conditions to minimize genetic variability. Thus, the observed differences in gene expression are likely reflective of phenotypic plasticity rather than genetic divergence.

To investigate this, a differential gene expression (DGE) analysis was performed for key detoxifying gene families, including CYP, CCE, CP, ABC, UGT, and GST, across three treatment groups: JD_NT (control, 0 ppm etofenprox), JD_25ppm (sublethal dose), and JD_100ppm (higher dose). The heat map revealed significant differences in gene expression among these groups (Figure 3).

The JD_25ppm group exhibited markedly higher DGE across all detoxification gene families compared to JD_NT and JD_100ppm. The rank order of DGE in the JD_25ppm group was CYP > CCE > CP > ABC > UGT > GST, suggesting that CYP and CCE families play a dominant role in the response to sublethal etofenprox exposure. In contrast, the JD_100ppm group demonstrated reduced expression compared to JD_25ppm, potentially due to metabolic saturation or distribution at higher concentrations, which may impair detoxification efficacy.

These findings highlight the plasticity of detoxification enzyme gene expression in WBPH populations from Jindo, particularly under sublethal etofenprox exposure. The elevated expression of CYP and CCE family’s genes in the JD_25ppm underscores their adaptive significance in mediating response at low insecticides concentrations. This study provides valuable insights into how detoxification pathways contribute to resistance development through phenotypic plasticity.

Further supporting our findings, a high degree of correlation was observed among replicates of JD_NT, JD_25ppm, and JD_100ppm (Figure 2), ensuring the reliability of the results. The consistency of these findings underscores the importance of sublethal does of etofenprox in modulating resistance mechanisms in WBPH populations. Understanding these adaptive transcriptional responses is critical for informing resistance management strategies and mitigating resistance development in pest populations exposed to sublethal insecticide concentrations.

### 2.8. Gene Expression Analysis of CYP and CCE Possibly Involved in Etofenprox Resistance

To validate the relationship between the expression of *CYP* and *CCE* genes and the resistance levels of WBPH to etofenprox, we quantified the fragment per kilobase of transcript per million mapped reads (FPKM) values. RNA-seq reads were mapped to the gene sequences to assess the expression profiles of genes from the CYP and CCE detoxification families at varying concentrations of etofenprox (0 ppm, 25 ppm, and 100 ppm). Our analysis focused on identifying the critical genes contributing to WBPH resistance under sublethal exposure to etofenprox (Figure 4 and Figure 5).

Our findings demonstrated a marked increase in the expression levels of specific *CYP* genes in response to etofenprox exposure. In particular, genes from the CYP6 family, including *CYPSF* and *CYP6AX3*, were highly upregulated across all treatments. Additionally, members of the CYP4 family, *CYP4G115* and *CYP4C62*, also exhibited increased expression under all tested concentrations of etofenprox in the Jindo WBPH population (Figure 4). This suggests a prominent role for these CYP genes in enhancing metabolic resistance to etofenprox.

Similarly, in the CCE gene family, a significant overexpression in the *CCE6-like* genes such as *XP 039294727.1 CCE-6* and *XP 022201494.2 CCE-6* genes was observed in the JD_25ppm group compared to the JD_NT and JD_100ppm groups. On the other hand, *E4-like* genes such as *AAG40239.1 E4-like* expression was higher in WBPH under all treatments of etofenprox. In addition, a significant increase was also observed in the JD_25ppm group compared to the JD_NT and JD_100ppm groups. These findings indicate a potential contribution of these two *CCE* genes to the detoxification process at sublethal concentrations of etofenprox. However, no considerable variation was detected in the expression of other CCE family genes in response to different etofenprox dosages, suggesting a more selective involvement of specific CCE genes in resistance mechanisms (Figure 5).

Overall, these results highlight the critical role of CYP6 and CYP4 family genes, along with selective *CCE* genes, in mediating WBPH resistance to etofenprox. The overexpression of these detoxification-related genes at sublethal doses provides valuable insights into the molecular mechanisms underpinning insecticide resistance in WBPH populations.

## 3. Discussion

Rice is a vital staple crop worldwide, but its productivity faces significant threats from insect pests, particularly the WBPH, which can cause severe damage to rice plants [1,30]. To mitigate this impact, various insecticides are employed; however, it is essential to identify which insecticides are most effective in enhancing WBPH mortality. Therefore, we conducted bioassays to evaluate the efficacy of five insecticides—etofenprox, fenobucarb, chlorpyrifos-methyl, phenthoate, and acephate—against WBPH populations across three Korean regions: Sinan, Hadong, and Jindo.

Our bioassay results showed that fenobucarb, chlorpyrifos-methyl, phenthoate, and acephate were highly effective in controlling the WBPH populations, achieving mortality rates exceeding 97% across all tested locations. In contrast, etofenprox exhibited significant variation depending on the region with much lower mortality rates recorded in Hadong (42.4 ± 5.2%), Sinan (40.0 ± 10.0%), and Jindo (27.3 ± 9.1%). These findings highlight the distinct regional differences in WBPH susceptibility to insecticides, particularly the Jindo population, which displayed notably high levels of resistance to etofenprox. This regional variability emphasizes the need for location-specific strategies in managing WBPH and highlights the challenges of relying solely on chemical control measures for this pest.

Our observations are consistent with prior studies that reported similar regional differences in pesticide efficacy against WBPH. For example, recent research found that insecticides such as etofenprox, fenobucarb, carbosulfan, dinotefuran, and imidacloprid exhibited moderate to high mortality (75–93%), while pymetrozine and buprofezin achieved lower mortality rates, under 45%, across multiple locations [6]. Comparable resistance patterns were documented in China, where WBPH exhibited variable resistance to nitenpyram, thiamethoxam, dinotefuran, clothianidin, chlorpyrifos, etofenprox, and isoprocarb, as reported by Huang et al. [31]. These findings collectively highlight the importance of understanding how different pesticides affect WBPH across diverse geographic regions to improve pest management efficacy.

To learn more about the resistance seen in the Jindo WBPH community, we examined the etofenprox LC_50_ value. Etofenprox alone was found to have an LC_50_ value of 241.7 ppm (95% C.I.: 157.69 to 373.28 ppm). Etofenprox’s LC_50_ value, however, dropped significantly to 51 ppm (95% C.I.: 1.06 to 14.47 ppm) when coupled with 0.5% PBO. The observed resistance in the Jindo WBPH population may be mostly attributed to metabolic resistance mechanisms, which are most likely mediated by detoxification enzymes, as indicated by the substantial reduction in LC_50_. These findings align with earlier research, including a study that identified GST and CYP as the main detoxifying enzymes responsible for buprofezin resistance in WBPH [19]. Their study showed a similar reduction in resistance with PBO, which further implicates these enzymes in the detoxification process.

We explored potential target-site mutations and metabolic resistance to further understand the drivers of etofenprox resistance. Our analysis revealed no mutations in the sodium channel genes (V253F, M918T, L1014F, and F1534C) in WBPH samples from all tested populations, suggesting that resistance mechanisms unrelated to sodium channel mutations are at play (Figure 1A,B). Instead, our findings corroborate previous research indicating a significant role for *CYP* genes in the development of resistance in *N. lugens* to etofenprox, supporting the hypothesis that metabolic resistance mechanisms underlie the observed resistance in WBPH [32]. Our findings also indicate that in addition to mutation, another mechanism of insecticide resistance, possibly metabolic resistance, contributes to the resistance of WBPH to etofenprox.

To investigate the sublethal effects of etofenprox, we conducted a transcriptomic analysis on Jindo populations exposed to different concentrations of etofenprox. Three groups were created: JD_NT (untreated and susceptible), JD_25ppm, and JD_100ppm. Our results showed that WBPH treated with 25 ppm etofenprox exhibited a significantly higher overexpression of genes compared to untreated controls and the 100 ppm treatment group (Figure 2). These results are consistent with a previous study that demonstrated that sublethal concentrations of sulfoxaflor significantly increased adult longevity and fecundity in WBPH [23], suggesting that sublethal effects may be a common response to certain insecticides, including etofenprox.

An enzyme mechanism for detoxifying the insecticides found in insects helps lessen pesticides’ harmful effects such as the development of resistance [33]. We investigated this by measuring the expression of detoxifying gene families such as CP, CCE, UGT, GST, ABC, and CYP. WBPH treated with 25 ppm of etofenprox exhibited a substantial upregulation of CYP and CCE gene families compared to untreated and 100 ppm-treated groups (Figure 3, Figure 4 and Figure 5). These findings are in line with the research of Chang et al. [34], who reported that the application of buprofezin on WBPH increased CCE activity, emphasizing the role of *CCE* genes in increasing the resistance of WBPH to various insecticides. Similarly, our findings align with previous research [31], highlighting the significant role of CYP enzymes, specifically through P450, in mediating resistance to etofenprox in *N. lugens*.

The elevated expression of detoxifying enzyme genes, particularly in response to 25 ppm of etofenprox, may come with a fitness cost, as maintaining high levels of CYP and CCE activity can impose a metabolic burden on WBPH. This potential trade-off highlights the importance of managing insecticide resistance carefully, as the associated fitness cost could impact the population sustainability under non-exposure conditions. The substantial reduction in LC_50_ from 241.7 to 51 upon the addition of PBO in our study further underscores the contribution of P450 monooxygenases to etofenprox detoxification, suggesting a synergistic potential of metabolic inhibition. Together, our results, in line with previous studies, reinforce that upregulated *CYP* and *CCE* genes play a critical role in etofenprox resistance in WBPH, providing insights into molecular mechanisms that could inform resistance management strategies in the field [33].

However, it is still difficult to answer why JD_NT and JD_25ppm show such significant differences in gene expression, while JD_100ppm indicates only a slight difference. We can only speculate because we have not yet found any explicit references or evidence for this part. We guess that if the reaction threshold is exceeded, metabolism is not activated. In other words, 25 ppm is a concentration sufficient to detect and react to foreign substances and activate metabolism, such as detoxification enzymes. However, activating this system at a high concentration of about 100 ppm is not easy, so this phenomenon seems to occur. Further research is ongoing, and a more comprehensive review will be possible once we understand detoxification enzyme activation and the entire detoxification system more thoroughly.

Despite the robustness of our approaches, including bioassay, DEG analysis using FPKM values, and mutation surveys, we acknowledge the absence of functional validation experiments in this study. While our findings provide preliminary insights into the molecular mechanisms of etofenprox resistance, future work should focus on validating the roles of specific detoxification genes through RNAi or CRISPR-based gene editing. Additionally, enzymatic activity assays and metabolic profiling could further elucidate the functional contributions of CYPs and CCEs to resistance. These steps will complement the current findings and provide a more comprehensive understanding of resistance mechanisms, ultimately informing sustainable pest strategies.

## 4. Materials and Methods

### 4.1. Insects

The white-backed planthopper (WBPH), *Sogatella furcifera* (Horváth) (Hemiptera: Delphacidae) field populations were collected from Hadong (34°58′23″ N 127°50′54″ E), Jindo (34°30′26″ N 126°16′57″ E), and Sinan (34°50′40″ N 126°21′40″ E) at 2019.

All WBPHs were maintained the same way with that BPH under controlled conditions at a temperature of 25  ±  1 °C, relative humidity of 60  ±  5%, and a photoperiod regime of 14 h of light and 10 h of darkness, as previously reported by Khan et al. [35].

### 4.2. Bioassay

Bioassays were conducted using the Insecticide Resistance Action Committee (IRAC) susceptibility test method 005, with some modifications (www.irac-online.org), for each WBPH adult. In our current study, we exposed WBPH to five different insecticides, namely etofenprox (3A), fenobucarb (1A), chlorpyrifos-methyl (1B), phenthoate (1B), and acephate (1B) across three regions of Korea (Sinan, Hadong, and Jindo) in 2019. Additionally, we exposed the WBPH population from Jindo to various concentrations (0 ppm, 25 ppm, and 100 ppm) of etofenprox to assess its sublethal effects. The untreated (0 ppm) Jindo WBPH population is denoted as JD_NT, while the Jindo WBPH populations treated with 25 and 100 ppm of etofenprox are denoted as JD_25 and JD_100, respectively.

After diluting the above-mentioned five insecticides to various concentrations, rice seedlings ten days after seeding were dipped into the diluted solution for 10 s, following which ten WBPH adults per treatment were transferred. All experiments were conducted with more than three biological replications (*n* > 30 per concentration), and further detailed bioassay methods were applied following the IRAC susceptibility test method 005.

Using the SAS program, based on the Probit model (SAS Institute 9.1, Cary, NC, USA), concentration-based mortality after two days of exposure to the above-mentioned five insecticides was estimated to determine the median lethal concentration (LC_50_) and 95% confidence intervals (CIs). The resistance ratio (RR) was calculated by dividing the LC_50_ value of the treated field population by that of the untreated field population.

### 4.3. RNA and DNA Extraction

Within 12 h of emergence, total RNAs were extracted from the adults in each population of WBPH with twenty adults serving as a biological replicate in each sample. The RNeasy Mini Kit (Qiagen, Hilden, Germany) was used for RNA extraction following the manufacturer’s instructions. Using an Agilent 2200 TapeStation (Agilent Technologies, Santa Clara, CA, USA), the RNA was verified and quantified, and the RNA integrity was verified by running samples on a 1% agarose gel using electrophoresis. We used the SuperiorScript III cDNA Synthesis Kit (Enzynomics, Daejeon, Republic of Korea) for the reverse transcription procedure. Before the upcoming studies, the generated cDNA and total RNA were kept at −70 °C. Additionally, within 12 h of emergence, genomic DNA (gDNA) was extracted from each WBPH adult within twenty adults as a biological replicate. This was performed using DNeasy Blood & Tissue (Qiagen) per the manufacturer’s instructions, and Nanodrop (Nanodrop Technologies, Wilmington, DE, USA) was used to quantify the results.

### 4.4. Mutation Survey

The cDNA and gDNA were subjected to PCR to examine the mutations in the sodium channel gene under particular thermal conditions. The ProFlex PCR System (ThermoFisher Scientific, Waltham, MA, USA) was used, along with KOD FX polymerase (Toyobo Life Science, Osaka, Japan), along with suitable primer combinations and PCR conditions. The used primer sets are listed in Appendix A. The chromatograms were examined for mutations, and the PCR products were directly sequenced (Macrogen, Seoul, Republic of Korea), adhering to the approach described previously [14].

### 4.5. RNA-Seq Analysis

Within 12 h of emergence, total RNAs were extracted from the adults of each WBPH population, each sample including twenty adults as a biological replicate. The RNeasy Mini Kit (Qiagen, Hilden, Germany) was used for RNA extraction under the manufacturer’s instructions. The RNA was validated and quantified using an Agilent 2200 TapeStation (Agilent Technologies, Santa Clara, CA, USA), and RNA integrity was confirmed by running samples on a 1% agarose gel using electrophoresis. The TruSeq RNA sample Prep Kit v2 (Illumina, San Diego, CA, USA) was utilized to construct RNA-seq libraries. The samples were sequenced using the TruSeq 3000/4000 SBS Kit v3 (Macrogen, Seoul, Republic of Korea) on the Hiseq4000 plDEGatform.

Trimmomatic v0.38 was first used to treat the nine RNA-seq raw sequences to exclude adapters and low-quality sequences (Q30) from the raw data [36]. Using FastQC v0.11.7 (http://www.bioinformatics.babraham.ac.uk/projects/fastqc), it was verified that the trimmed reads produced had a good quality.

### 4.6. Clean Read Assembly and Unigene Construction

To create transcriptomic references, trimmed reads from each sample were merged into an assembly group. The Trinity version (r20140717) program, which is used for de novo transcriptome assembly, was used to put the merged data together [37]. Contigs, or transcript fragments, are the end product of this procedure. Using the CD-HIT-EST tool made available by CD-HIT v4.6, the longest contigs were clustered into non-redundant transcripts known as unigenes [38].

### 4.7. Functional Annotation

Using TransDecoder v3.0.1, protein sequences were produced by predicting ORFs from unigenes. Annotation and expression analysis were conducted using the resulting unigenes and protein sequences. For functional annotation, we utilized BLASTN from NCBI BLAST v2.9.0+ and BLASTX of DIAMOND v0.9.21 software. We used the default E-value cutoff of 1.0 × 10^−5^ [39]. The databases were searched against the Kyoto Encyclopedia of Genes and Genomes (KEGG), NCBI Nucleotide (NT), Pfam, Gene Ontology (GO), NCBI non-redundant Protein (NR), UniProt, and EggNOG [40].

Using Bowtie v1.1.2, trimmed reads from each sample were aligned to the assembled unigene as a reference. In the context of differentially expressed gene analysis, the RSEM v1.3.1 method estimated the abundances of unigenes across samples into the read count as an expression measure [41]. For nine samples, if more than one read count value was 0, it was not included in the analysis. We used DESeq2 v1.28.1 (https://www.bioconductor.org/packages/release/bioc/html/DESeq2.html) to use Relative Log Expression (RLE) normalization and estimate the size factors from the count data to minimize systematic bias. The statistical analysis and analysis of differentially expressed genes (DEGs) were carried out with Log2FoldChange (FC) and nbinomWaldTest per comparison pair using DESeq2 (|FC| > 2 and nbinomWaldTest raw *p*-value < 0.05).

## 5. Conclusions

This study examines the responses of WBPH populations from various regions of Korea to five commonly used insecticides. While fenobucarb, chlorpyri-fos-methyl, phenthoate, and acephate demonstrated high efficacy, achieving over 97% mortality rates across all populations, etofenprox revealed significant regional variability. Notably, the Jindo population exhibited the highest resistance to etofenprox, resulting in a considerably lower mortality rate than the Sinan and Hadong populations. Our analysis of the LC_50_ values for etofenprox confirmed that metabolic resistance mechanisms, likely driven by detoxification processes, are critical to this resistance. Adding PBO significantly reduced the LC_50_ for etofenprox, suggesting the involvement of enzymes such as CYP in the detoxification process. The transcriptomic analysis supports these findings, revealing the upregulation of detoxification genes, including *CYPs* such as *CYPSF*. Importantly, no mutations were found in sodium channel genes associated with target-site resistance, indicating that alternative mechanisms, specifically metabolic pathways, are responsible for the observed resistance. These results underline the complexity of insecticide resistance in WBPH and the need for region-specific pest management strategies. Our findings emphasize the importance of monitoring insecticide resistance and adapting pest control approaches based on local population responses. Further research, including quantitative PCR validation, is needed to confirm the molecular mechanisms underlying this resistance and to develop more effective integrated pest management strategies for controlling WBPH in rice cultivation.

## Figures and Tables

**Figure 1 ijms-25-13605-f001:**
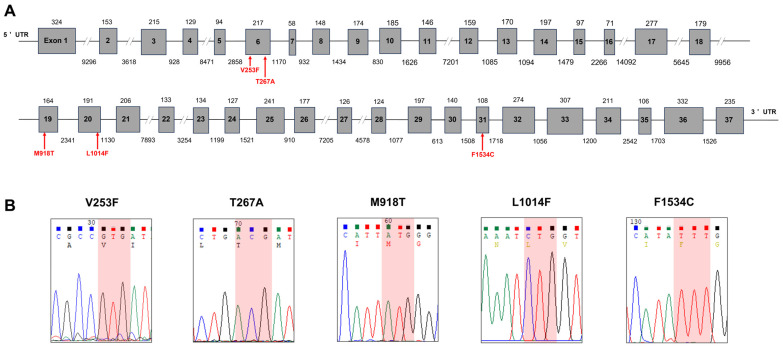
Mutation survey results of the voltage-gated sodium channel (VGSC), a well-known mechanism of resistance to pyrethroid insecticides such as etofenprox in *Sogatella furcifera*. (**A**). The full-length VGSC gene sequence of *S. furcifera* was elucidated (PQ306230, 6422 bp), and five previously reported mutation sites were confirmed. The VGSC gene, which encodes 2069 amino acids, consists of 37 exons based on the reference genome (GCA_017141385.1). The five mutation sites are in exons 6, 19, 20, and 31, respectively. (**B**). The five previously reported mutations were absent and confirmed by Sanger sequencing. Pooled gDNA from 20 adults of the Jindo population was used for the mutation survey.

**Figure 2 ijms-25-13605-f002:**
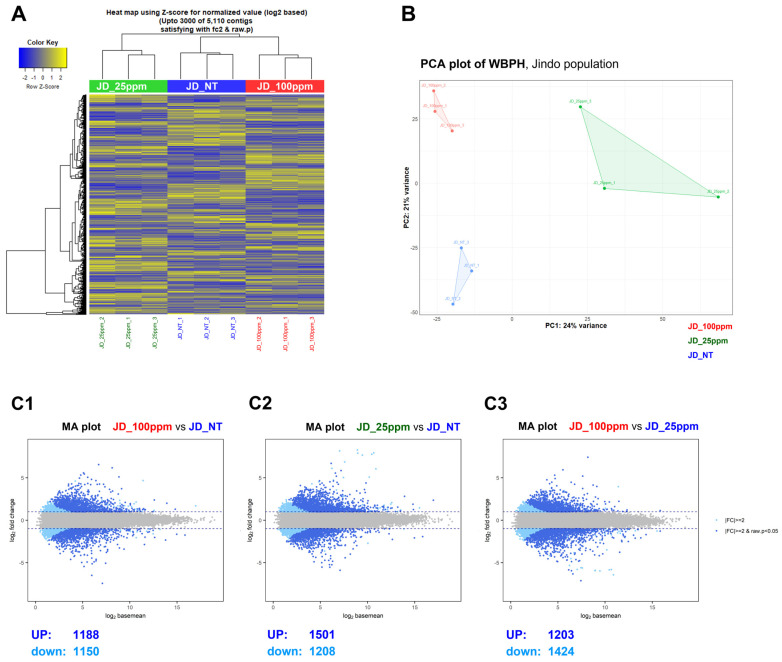
Differential gene expression analysis results. (**A**) Hierarchical clustering using Z-score for normalized gene expression values (log2 based) across JD_NT, JD_25ppm, and JD_100ppm of WBPH populations at 0, 25, and 100 ppm concentrations of etofenprox on WBPH populations from Jindo. (**B**) Principal component analysis (PCA) displaying the genetic relationships among JD_NT, JD_25ppm, and JD_100ppm of WBPH populations from Jindo. (**C1**–**C3**) Mean average (MA) plots comparing gene expression among JD_NT, JD_25ppm, and JD_100ppm of WBPH populations from Jindo. The region that exists in the part where the gray dots are located is the part that has no statistically significant expression difference. The differently expressed genes are marked in blue (UP) and sky blue (down).

**Figure 3 ijms-25-13605-f003:**
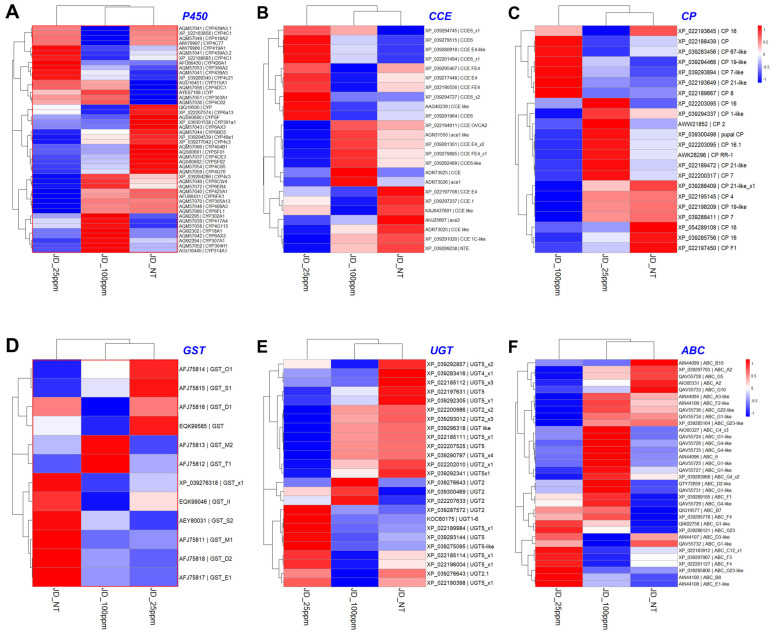
Heat map showing the expression of six major detoxification enzyme genes. (**A**–**F**) Cytochrome P450 (CYP), carboxyl/cholinesterase (CCE), cuticular protein (CP), glutathione S-transferase (GST), 5′-diphosphate-glucosyltransferase (UGT), and ATP-binding cassette transporter (ABC), respectively.

**Figure 4 ijms-25-13605-f004:**
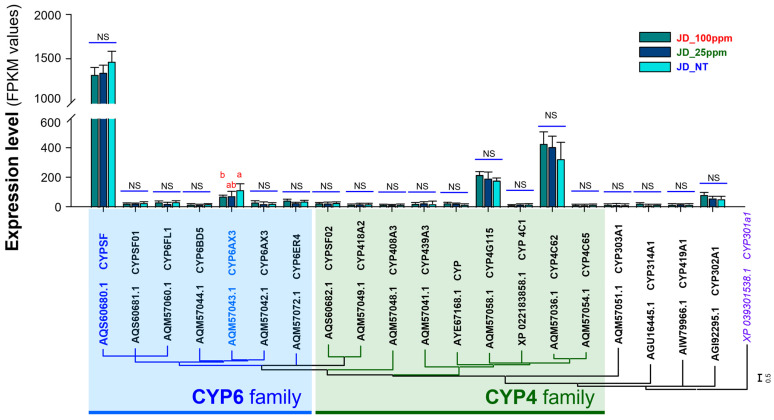
Expression comparison and phylogenetic analysis of the top 20 *CYP* genes with high expression levels. The expression levels were compared based on fragment per kilobase transcript per million mapped reads (FPKM) values, and the “a, b, ab” indicate significant differences (Tukey’s multiple comparison test, *p* < 0.01), while “NS” indicate non-significant differences. The phylogenetic analysis of genes was performed using MEGA11, and the analysis was performed using the neighbor-joining method at the amino acid level. Mitochondrial *CYP301a1* was used as an out-group.

**Figure 5 ijms-25-13605-f005:**
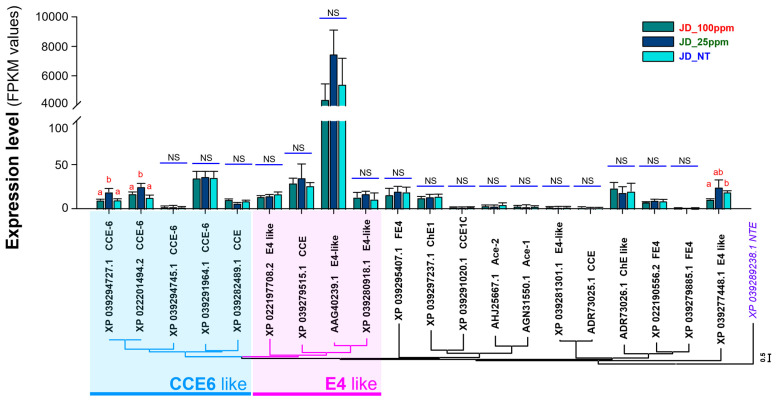
Expression comparison and phylogenetic analysis of the top 20 CCE genes with high expression levels. The expression levels were compared based on fragment per kilobase transcript per million mapped reads (FPKM) values, and the “a, b, ab” indicate significant differences (Tukey’s multiple comparison test, *p* < 0.01), while “NS” indicate non-significant differences. The phylogenetic analysis of genes was performed using MEGA11, and the analysis was performed using the neighbor-joining method at the amino acid level. NTE (neuropathy target esterase) was used as an out-group.

**Table 1 ijms-25-13605-t001:** Bioassay results for three field populations of WBPH after exposure to five different insecticides.

Insecticides (MoA) *	Mortality (%)		
	2019 Sinan	2019 Hadong	2019 Jindo
Etofenprox (3A)	40.0 ± 10.0	42.4 ± 5.2	27.3 ± 9.1
Fenobucarb (1A)	100.0 ± 0.0	100.0 ± 0.0	100.0 ± 0.0
Chlorpyrifos-methyl (1B)	100.0 ± 0.0	100.0 ± 0.0	100.0 ± 0.0
Phenthoate (1B)	100.0 ± 0.0	97.0 ± 5.2	100.0 ± 0.0
Acephate (1B)	100.0 ± 0.0	100.0 ± 0.0	100.0 ± 0.0

(MoA) * stands for Mode of Action and is a classification of insecticides by the Insecticide Resistance Action Committee (IRAC) based on their mechanism of action. For more information, please visit the link below: https://irac-online.org/ (accessed on 1 October 2024).

**Table 2 ijms-25-13605-t002:** Synergistic effect of cytochrome P450.

2019 Jindo	LC_50_ Values (mg/L) (95% CI)	Slope	χ2 Log10 (Dose)	SR *
Etofenprox	241.7 (157.7–373.3)	1.15	3.42	4.7
Etofenprox + 0.5% PBO	51.0 (10.6–144.7)	0.58	1.81	

SR * (synergistic ratio) = LC_50_ of Etofenprox/LC_50_ of Etofenprox + 0.5% PBO (piperonyl butoxide, cytochrome P450 inhibitor).

**Table 3 ijms-25-13605-t003:** Statistics of initial assembled, and unigene contig.

Assembly	No. of Genes	No. of Transcripts	GC (%)	N_50_	Avg. Contig Length (bp)	Total Assembled Bases (bp)
Initial assembled contig	222,775	294,336	39.82	1330	745.45	219,415,085
Unigene contig	194,501	222,775	39.38	1080	660.27	128,423,880

**Table 4 ijms-25-13605-t004:** Statistics of ORF prediction.

Assembly	Total Unigene	ORF Predicted Unigene	Single ORF Predicted Unigene	Multiple ORF Predicted Unigene
Merge	194,501	27,079 (13.92%)	24,705 (91.23%)	2374 (8.77%)

## Data Availability

The datasets generated during the current study are available from the corresponding author upon reasonable request.

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
