# Peer review of "Plasticity in Gene Expression Patterns and CYPSF Gene Possibly Involved in the Etofenprox-Resistant Population of White-Backed Planthopper, Sogatella furcifera"

_ijms, 2024, doi:10.3390/ijms252413605_

Round 1

Reviewer 1 Report

Comments and Suggestions for Authors

This work aimed to identify genes potentially associated with etofenprox resistance in Sogatella furcifera. The authors performed a bioassay to identify the most etofenprox-resistant field population.  Once the field population with the highest sensitivity to etofenprox was identified, the respective procedures were carried out to investigate molecular mechanisms of resistance such as voltage-gated sodium channel mutations by complete sodium channel sequencing and identifying candidate resistance genes. The sensitive population was subdivided to expose the organisms to two doses of etofenprox and subsequently, by RNA-seq, differences in gene expression in general were determined, and DGE for CYP, CCE, CP, ABC, UGT and GST gene families. Potential mutant sites in the nucleotide sequence the sodium channel that produce resistance to pesticide in insect were located, but changes were not observed WBPH. Differences in gene expression in general were observed. The 25-ppm dose showed significant differences with NT, while NT did not differ in gene expression with 100 ppm. The greatest differences in DEG were observed in organisms exposed to 25 ppm of etofenprox.

The study is very interesting and presents very valuable information for understanding the molecular mechanisms of resistance through detoxification. There are one element that I think is very important to review and clarify, is regarding plasticity. I think it is important to mention in the introduction whether the authors consider it as a mechanism of resistance to insecticides and especially how they define plasticity for the study, since it seems to me that it can confuse the reader interested in the evolution of plasticity as a characteristic or byproduct of natural selection (see DeWitt and Scheiner 2004, phenotypical plasticity, functional and conceptual approaches, Oxford University Press).

It is also important to ensure that the complementary tables and figures are available. I did not find access to them and the information could not be checked.

Below are some specific observations

Line 82, I think a few lines are needed to say why it is important to demonstrate resistance to etofenprox

95, Is it correct that the last three insecticides have 1B associated with them? If so, what do the numbers and letters in parentheses mean? Also, check that there is congruence between the figure caption and the information given in the text, in the current presentation they are not completely congruent.

00-101. What result does this article show "no WBPH-susceptible strains to etofenprox?", I am not clear on the relevance of the idea, it seems out of place. Perhaps the idea is "this result does not allow evaluating resistance since it does not include lines with different susceptibility to etofenprox."

104, delete “further”

119, I suggest that you indicate what the numbers and literals in parentheses in the first column mean.

121, add the probability of Chi-square, and explain what SR means

154-158- Please review this sentence, is a bit confusing. The sentence "To ensure the reliability of RNA-seq data for gene expression analysis, sequencing was performed on three distinct WBPH populations..." seems somewhat inconsistent with the goals of estimating resistance and molecular mechanisms of resistance. Is the dose assay really intended to ensure that the RNA-seq information is good or true? Or did the authors use RNA-seq to explore molecular mechanisms of resistance, for example gene overexpression? I assume they expected the level of expression to correlate positively with etofenprox doses. I consider that reliability of the RNA-seq was assessed by the three biological replicates.

204-206, It is the same information or the same data, so the idea is confusing or ambiguous.

251, what is MA plot

280, I find it very interesting to include the perspective of plasticity. But it causes me some uncertainty and I have doubts, since no background is presented in the introduction. Phenotypic plasticity, which in a strict sense has been defined as the capacity of a genotype to express different phenotypes depending on an environmental gradient (DeWitt and Scheiner 2004). Here the differential expression of genes in three subpopulations was studied, however it has not been said whether the populations subjected to different doses have the same genotype. It is advisable to briefly explain the perspective of plasticity used in this study, so that the reader can assess the study in the general context of the study of phenotypic plasticity.

424, Please indicate the relationship between etofenprox and sulfoxaflor…

457, I suggest that the authors discuss further why 25 ppm differs from NT, and why NT and 100 ppm does not differ, those results seem very strange or counterintuitive to me.

Author Response

Reviewer 1.

This work aimed to identify genes potentially associated with etofenprox resistance in Sogatella furcifera. The authors performed a bioassay to identify the most etofenprox-resistant field population.  Once the field population with the highest sensitivity to etofenprox was identified, the respective procedures were carried out to investigate molecular mechanisms of resistance such as voltage-gated sodium channel mutations by complete sodium channel sequencing and identifying candidate resistance genes. The sensitive population was subdivided to expose the organisms to two doses of etofenprox and subsequently, by RNA-seq, differences in gene expression in general were determined, and DGE for CYP, CCE, CP, ABC, UGT and GST gene families. Potential mutant sites in the nucleotide sequence the sodium channel that produce resistance to pesticide in insect were located, but changes were not observed WBPH. Differences in gene expression in general were observed. The 25-ppm dose showed significant differences with NT, while NT did not differ in gene expression with 100 ppm. The greatest differences in DEG were observed in organisms exposed to 25 ppm of etofenprox.

Comments:

The study is very interesting and presents very valuable information for understanding the molecular mechanisms of resistance through detoxification. There are one element that I think is very important to review and clarify, is regarding plasticity. I think it is important to mention in the introduction whether the authors consider it as a mechanism of resistance to insecticides and especially how they define plasticity for the study, since it seems to me that it can confuse the reader interested in the evolution of plasticity as a characteristic or byproduct of natural selection (see DeWitt and Scheiner 2004, phenotypical plasticity, functional and conceptual approaches, Oxford University Press).

Response:

We sincerely appreciate your time, thoughtful comments, and valuable suggestions, which will improve the quality and clarity of our manuscript. Based on your valuable comments and suggestions we have revised our manuscript using the track changes tool.

We appreciate your insightful comment regarding the need to clarify the use of plasticity in the context of our study. In response, we have revised the introduction to define plasticity as it relates to our findings explicitly. In this study, we use plasticity to refer to the reversible, dynamic changes in gene expression that occur in response to environmental stimuli, such as insecticide exposure. These changes do not involve permanent genetic mutations but reflect an organism's ability to adapt metabolically through altered expression of detoxification-related genes, such as CYPs, GSTs, and UGTs.

To prevent confusion, we have distinguished this functional plasticity from the evolutionary concept of plasticity as a heritable trait shaped by natural selection, as discussed in DeWitt and Scheiner (2004). The revised introduction and result sections (Kindly see lines 94-100 and 302-360) now clearly delineate this context, ensuring it aligns with the study's focus on metabolic resistance mechanisms rather than evolutionary plasticity.

Comments:

It is also important to ensure that the complementary tables and figures are available. I did not find access to them and the information could not be checked.

Response:

When submitting the manuscript draft, we uploaded supplementary data as a separate file, but this problem probably occurred due to a technical error or our mistake. We have now ensured that all complementary tables and figures are included as supplementary files with the submission. These materials contain detailed data on various analysis related to the manuscript. We have also cross-referenced these materials within the main manuscript text to improve accessibility and ensure readers can easily locate and interpret the additional information.

Below are some specific observations

Comment: Line 82, I think a few lines are needed to say why it is important to demonstrate resistance to etofenprox

Response:

Thank you so much for your valuable suggestion. We have added the suggested information on the revised manuscript from lines 84-86.

Comment: 95, Is it correct that the last three insecticides have 1B associated with them? If so, what do the numbers and letters in parentheses mean? Also, check that there is congruence between the figure caption and the information given in the text, in the current presentation they are not completely congruent.

Response:
Thank you for pointing this out. The classification (1A and 1B) represents the insecticide's mode of action (MoA) group as per the Insecticide Resistance Action Committee (IRAC) guidelines. 1A refers to carbamates, which inhibit acetylcholinesterase. In comparison, 1B refers to organophosphates, which also act as acetylcholinesterase inhibitors but belong to a different chemical class. Please see lines 106-109 of the revised manuscript. Thank you. Furthermore, in this section (2.1) of the results, we have only mentioned and discussed the tables, not the figures.

Comment: 00-101. What result does this article show "no WBPH-susceptible strains to etofenprox?", I am not clear on the relevance of the idea, it seems out of place. Perhaps the idea is "this result does not allow evaluating resistance since it does not include lines with different susceptibility to etofenprox."

Response:
Thank you so much for your meticulous point. We have revised the sentence to avoid any confusion. Please see lines 113-114 of the revised manuscript.

Comment: 104, delete “further”

Response:
Deleted, thank you. Please see line 118 of the revised manuscript.

Comment: 119, I suggest that you indicate what the numbers and literals in parentheses in the first column mean.

Response:

Thank you for your detailed advice, we believe that it would enhance the clarity of our manuscript and will make it reader-friendly. We added the relevant information as a comment to Table 1. Please see lines 134-136 of the revised manuscript.

Comment: 121, add the probability of Chi-square, and explain what SR means

Response:

Thank you so much for pointing out these important points. We have edited Table 2 as you advised. Please see line 138 of the revised manuscript.

Comment: 154-158- Please review this sentence, is a bit confusing. The sentence "To ensure the reliability of RNA-seq data for gene expression analysis, sequencing was performed on three distinct WBPH populations..." seems somewhat inconsistent with the goals of estimating resistance and molecular mechanisms of resistance. Is the dose assay really intended to ensure that the RNA-seq information is good or true? Or did the authors use RNA-seq to explore molecular mechanisms of resistance, for example gene overexpression? I assume they expected the level of expression to correlate positively with etofenprox doses. I consider that reliability of the RNA-seq was assessed by the three biological replicates.

Response:

We have revised the sentence to avoid confusion. Kindly see lines 168-172 of the revised manuscript. Thank you.

204-206, It is the same information or the same data, so the idea is confusing or ambiguous.

Response:

We have revised the sentence to avoid confusion. Kindly see lines 215-229 of the revised manuscript. Thank you.

Comment: 251, what is MA plot

Response:

We have added the information regarding the MA plot in the revised manuscript. Kindly see lines 266-274 of the revised manuscript. Thank you.

Comment: 280, I find it very interesting to include the perspective of plasticity. But it causes me some uncertainty and I have doubts, since no background is presented in the introduction. Phenotypic plasticity, which in a strict sense has been defined as the capacity of a genotype to express different phenotypes depending on an environmental gradient (DeWitt and Scheiner 2004). Here the differential expression of genes in three subpopulations was studied, however it has not been said whether the populations subjected to different doses have the same genotype. It is advisable to briefly explain the perspective of plasticity used in this study, so that the reader can assess the study in the general context of the study of phenotypic plasticity.

Response:

We have included the suggested information regarding the plasticity in both the introduction and result sections of the manuscript and we hope that it will enhance the clarity of our manuscript and will make it reader-friendly. Kindly see lines 94-100 and 302-360 of the revised manuscript. Thank you.

Comment: 424, Please indicate the relationship between etofenprox and sulfoxaflor…

Response:

Thank you for your comment. We accept that there is no direct relationship between etofenprox and sulfoxaflor. Therefore, in the revised manuscript we have revised the sentence to clarify that the consistency lies in the observed sublethal effects on WBPH rather than a relationship between the two insecticides. Kindly see lines 474-478 of the revised manuscript.

Comment: 457, I suggest that the authors discuss further why 25 ppm differs from NT, and why NT and 100 ppm does not differ, those results seem very strange or counterintuitive to me.

Response:

We can only make assumptions because we haven't yet found any explicit references for this part. We guess that if the reaction threshold is exceeded, metabolism is not activated.

At 25 ppm, it is a concentration sufficient to detect and react to foreign substances and activate metabolism, such as detoxification enzymes. However, at a high concentration of about 100 ppm, it is difficult to activate this system, so this phenomenon seems to occur.

Further research is ongoing, and a more comprehensive review will be possible once we understand more about detoxification enzyme activation and the entire detoxification system.

Kindly see lines 508-517 of the revised manuscript.

Reviewer 2 Report

Comments and Suggestions for Authors

This manuscript reports the results of investigating Jindo population with strong resistance to the insecticide etofenprox and exploring the genetic factor that is involved in such a resistance using transcriptomic analysis. It is a very interesting research topic, and the research findings has the potential leading to an alternative strategy for pest management. The research approach is reasonable but needs to be expanded. The results from the experiments are preliminary but encouraging as some differentially expressed genes were identified, which was believed to relate to the phenotype of etofenprox resistance in white-backed planthopper (WBPH). Although the expression studies showed the findings of the possible connection between gene expression and resistance to etofenprox, the main concern is the lack of functional study of the explored genes. Validation of gene expression from RNA-seq data in one of the methods such as qRT-PCR or transgenic approach is necessary. The rationale here is that if you get the same results in two separate data sets, then the results are likely true no matter what technology is used. If the author could test the selected genes functionally, then the manuscript can be considered for publication in this esteemed journal.

In addition, there are several specific comments as shown below:

1.            Please clearly indicate each author’s email address as the current list is a little confused.

2.            qRT-PCR validation experiment is missing in the Material and Methods section.

3.            L123, correct the number of the subsection as it is not in numerical order.

4.            L515, should be italic.

5.            Keep the rules for capitalization in all titles of subsections such as L154, L172...

6.            Author may consider moving Figure  2C1-3  into Supplementary materials.

Author Response

Reviewer 2

This manuscript reports the results of investigating Jindo population with strong resistance to the insecticide etofenprox and exploring the genetic factor that is involved in such a resistance using transcriptomic analysis. It is a very interesting research topic, and the research findings has the potential leading to an alternative strategy for pest management. The research approach is reasonable but needs to be expanded. The results from the experiments are preliminary but encouraging as some differentially expressed genes were identified, which was believed to relate to the phenotype of etofenprox resistance in white-backed planthopper (WBPH). Although the expression studies showed the findings of the possible connection between gene expression and resistance to etofenprox, the main concern is the lack of functional study of the explored genes. Validation of gene expression from RNA-seq data in one of the methods such as qRT-PCR or transgenic approach is necessary. The rationale here is that if you get the same results in two separate data sets, then the results are likely true no matter what technology is used. If the author could test the selected genes functionally, then the manuscript can be considered for publication in this esteemed journal.

Response:

Thank you for your valuable comments and constructive suggestions. We appreciate your recognition of the importance and potential of our research on etofenprox resistance in white-backed planthopper (WBPH). We have addressed your main concerns and revised the manuscript by using the Tack Change tool.

  • Lack of functional validation in the current study:

We also agree with your meticulous point regarding the importance of functional validation for confirmation of key roles of the candidate genes associated with etofenprox resistance. Therefore, in the present study, we focused on the identification of differentially expressed genes (DEGs) and their pivotal roles through RNA-seq and FPKM-based expression analyses. Furthermore, we did not conduct qRT-PCR or transgenic studies due to time and available resources at our laboratory. However, our present study incorporated multiple robust approaches, including bioassays, mutation surveys, and DEG analysis, to support the RNA-seq findings.

In the current study, we recognized differential expression patterns in key detoxification-related genes, such as cytochrome P450s (CYPs) and carboxylesterases (CCEs), which were steadily observed across etofenprox-resistant populations. These results advocate their impending participation in resistance mechanisms. To reinforce future work, we plan to include qRT-PCR validation for selected candidate genes and, where practicable, employ RNAi or transgenic methodologies to functionally describe their roles in etofenprox resistance. This extended method will offer surplus validation of our results.

  • Preliminary results of the current study:

We appreciate your scientific approach regarding the findings of our current study. In the present study, we reported on the primary identification of DEGs related to etofenprox resistance, we trust it will provide a strong foundation for further functional studies. Our bioassays, genetic surveys, and RNA-seq analysis allow us to propose hypotheses about the genetic and metabolic factors contributing to resistance.

  • Future studies:

We recognize the significance of addressing the functional roles of recognized genes to solidify our conclusions. Therefore, we have revised the discussion section to underscore the limitations of the present study and outline our plans to validate the identified DEGs through qRT-PCR and functional assays. These follow-up studies will provide a more comprehensive understanding of the genetic mechanisms underlying etofenprox resistance. Please see lines 505-514 of the revised manuscript.

In addition, there are several specific comments as shown below:

  1. Please clearly indicate each author’s email address as the current list is a little confused.

Response 1:

We sincerely thank the reviewer for highlighting this important point to avoid any confusion regarding the authors' email addresses. To address this, we have listed each author's name alongside their respective email address. This clarification has been incorporated in lines 8–18 of the revised manuscript. We hope this amendment resolves any ambiguity. Thank you for bringing this to our attention.  

  1. qRT-PCR validation experiment is missing in the Material and Methods section.

Response 2:

We appreciate the reviewer’s comment regarding the qRT-PCR validation experiment. Although RNA-seq is re-confirmed through qPCR in some cases, it was not performed in this study because it was judged not necessary for experiments with the same purpose of expression comparison. Consequently, it has not been included in the Materials and Methods section of the manuscript. We thank the reviewer for understanding this aspect of the study's scope.

  1. L123, correct the number of the subsection as it is not in numerical order.

Response 3:

Thank you for pointing out the error in the numbering of the subsection at lines 137, 167, 186, 214, and 234 (in the revised manuscript). We have corrected the numbering to ensure it follows the proper numerical order. Additionally, we have carefully reviewed the entire manuscript to ensure consistency and accuracy in the section and subsection numbering throughout. We appreciate your attention to detail.

  1. L515, should be italic.
  2. Keep the rules for capitalization in all titles of subsections such as L154, L172...

Response 4 and 5:

Thank you for highlighting this formatting issue. We have corrected the text at line 515 by italicizing it as appropriate. Additionally, we have thoroughly reviewed the entire manuscript to ensure consistency in the use of italics where required. We appreciate your careful observation.

  1. Author may consider moving Figure 2C1-3 into Supplementary materials.

Response 6:

We appreciate your crucial points regarding our manuscript. However, we trust it is necessary to retain panels C1, C2, and C3 as part of the main Figure 2 rather than moving them to the supplementary materials. These panels provide significant perceptions into gene regulation patterns, as presented through MA plots for comparison between JD_100ppm vs JD_NT, JD_25ppm vs JD_NT, and JD_100ppm vs JD_25ppm.

For instance:

  • JD_100ppm vs JD_NT: 1,188 are upregulated, while 1,150 are downregulated.
  • JD_25ppm vs JD_NT: 1,501 are upregulated, while 1,208 are downregulated.
  • JD_100ppm vs JD_25ppm: 1,203 are upregulated, while 1,424 are downregulated.

These comparisons, as illustrated in Figure 2C1-3 (on line 295 of the revised manuscript), are crucial for understanding the magnitude and direction of gene expression changes under different treatment conditions, which is the main objective of our study. Therefore, with due respect, we suggest keeping these panels in the main manuscript to ensure clarity and accessibility of this key information for readers.

Round 2

Reviewer 2 Report

Comments and Suggestions for Authors

Authors have made some corrections and rewriting for improvement of the manuscript during the revision but failed to satisfy addressing the important comments on validation of identified DEG. In author’s response, they indicated that ”Although RNA-seq is re-confirmed through qPCR in some cases, it was not performed in this study because it was judged not necessary for experiments with the same purpose of expression comparison”. I would like to argue that validation of DEGs identified from RNA-seq analysis is crucial because the whole process of RNA-seq analysis can generate false positives; therefore, independent experimental verification through commonly used methods like qPCR or transgenic technology is necessary to confirm the true values and ensure reliable results for downstream analysis and interpretation, such as in the case of declaring a gene involved in such biological functions. Convincing evidence in manuscript is necessary to make people to believe whether the research is scientifically valid and technically sound to meet the standard for publication in those esteemed journals like this one, International Journal of Molecular Sciences.

Author Response

As the reviewer thinks, it is natural that the study's design and experimental verification should be in line with the journal's level. I also tried my best to make this study suitable for the International Journal of Molecular Sciences level.
However, as the reviewer pointed out, there may be differences of opinion about verifying RNAseq results with qPCR. Verification is necessary when RNAseq was not technically proven in the past or was performed without replications. However, this study performed biological replication in triplicate and presented statistically detailed analysis results.
I believe the purpose of performing RNAseq and qPCR is to compare gene expression. Then, whether qPCR should be performed again simply for technical confirmation is questionable.
Since research based on RNAseq results is currently being conducted in many different fields of biology, I think the reviewer is well aware that even prestigious journals accept RNAseq results alone. Many papers published in IJMS also sufficiently verify RNAseq results alone as expression analysis results.

Int. J. Mol. Sci. 2019, 20, 783; doi:10.3390/ijms20030783
Int. J. Mol. Sci. 2024, 25, 12963. https://doi.org/10.3390/ijms252312963
Int. J. Mol. Sci. 2023, 24(11), 9775; https://doi.org/10.3390/ijms24119775

Round 3

Reviewer 2 Report

Comments and Suggestions for Authors

Dear Authors, I believe that validation of DEGs identified from RNA-seq analysis is crucial and necessary to make sure that the research report is scientifically valid, which would demonstrate high quality of your publication and will help the journal maintain its standard of publication.